# Silver Nanoparticle-Mediated Synthesis of Fluorescent Thiolated Gold Nanoclusters

**DOI:** 10.3390/nano11112835

**Published:** 2021-10-25

**Authors:** Cheng-Yeh Chang, Tzu-Hsien Tseng, Bo-Ru Chen, Yi-Ru Wu, Cheng-Liang Huang, Jui-Chang Chen

**Affiliations:** Department of Applied Chemistry, National Chiayi University, Chiayi City 600355, Taiwan; sss1072727@gmail.com (C.-Y.C.); tzuhome50@gmail.com.tw (T.-H.T.); s1090251@mail.ncyu.edu.tw (B.-R.C.); jerry19970228@gmail.com (Y.-R.W.); clhuang@mail.ncyu.edu.tw (C.-L.H.)

**Keywords:** electrophoretic mobility, quantum yield, capping ligands, nanoparticles, nanoclusters

## Abstract

A new strategy using silver nanoparticles (Ag NPs) to synthesize thiolated Au NCs is demonstrated. The quasi-spherical Ag NPs serve as a platform, functioning as a reducing agent for Au (III) and attracting capping ligands to the surface of the Ag NPs. Glutathione disulfide (GSSG) and dithiothreitol (DTT) were used as capping ligands to synthesize thiolated Au NCs (glutathione-Au NCs and DTT-Au NCs). The glutathione-Au NCs and DTT-Au NCs showed red color luminance with similar emission wavelengths (630 nm) at an excitation wavelength of 354 nm. The quantum yields of the glutathione-Au NCs and DTT-Au NCs were measured to be 7.3% and 7.0%, respectively. An electrophoretic mobility assay showed that the glutathione-Au NCs moved toward the anode, while the DTT-Au NCs were not mobile under the electric field, suggesting that the total net charge of the thiolated Au NCs is determined by the charges on the capping ligands. The detection of the K_SV_ values, 26 M^−1^ and 0 M^−1^, respectively, revealed that glutathione-Au NCs are much more accessible to an aqueous environment than DTT-Au NCs.

## 1. Introduction

The nanoscience of noble metals has attracted a great deal of attention and become a promising research field in past decades. Among the noble metals, gold is one of the most extensively studied materials because of its chemical stability, nontoxicity, and biocompatibility. Gold displays characteristic properties that are dependent on its sizes. When processed into 100 nm or less in diameters, termed “nanoparticles” (NPs), Au shows unique properties distinct from bulk Au. These gold nanomaterials are further categorized into two classes based on their dimensions and properties: gold nanoparticles (Au NPs) and gold nanoclusters (Au NCs). Gold nanoclusters are particles typically less than 3 nm in diameter and are composed of several to tens of gold atoms. The ultra-small nanoparticles are in a league of their own as their distinct properties are not found in Au nanoparticles. Unlike Au NPs, Au NCs neither appear in a visible geometrical shape under TEM or SEM, nor do they possess surface plasmon resonance (SPR) absorption. Consequently, Au NCs show distinct optical properties when compared to those of Au NPs and bulk gold. Au NPs absorb visible light while Au NCs do not. The most important characteristics setting Au NCs apart from nanoparticles is that Au NCs exhibit a molecular-like luminescence property [1,2,3]. The size of these particles is comparable to an electron Fermi wavelength of gold metal, and its continuous band structure breaks into discrete energy states [1]. As a result, these Au NPs behave like molecules and emit fluorescence [4,5]. Multicolored fluorescent gold clusters with different sizes have been synthesized and the results indicate the size-dependent emission of Au NCs [6]. In addition, the report also reveals that the optical properties of Au NCs are highly dependent on their sizes, structures, oxidation states, and capping ligands, as well as on environmental parameters, such as temperature, pH, and ionic strength [7].

The synthesis of Au nanoclusters is generally classified into two approaches: “nanoparticles to clusters” and “atoms to clusters”. Both approaches require ligand stabilizers or capping ligands during synthesis [7]. With conventional Au NPs serving as a starting material, the “nanoparticles to clusters” method typically involves two tedious steps. This approach has been systematically studied and a mechanism has been proposed [8]. Nevertheless, this approach is time-consuming, and isolation is usually required. In the “atoms to clusters” approach, Au clusters are formed from the reduction of Au ions into atoms, followed by the nucleation of the Au atoms. However, Au (III) could be reduced into Au nanoparticles easier than into nanoclusters because of the potential of Au atoms to aggregate. Therefore, appropriate capping ligands are required for the synthesis of finite small-size clusters. A previous result has shown that stable Au clusters were generated by hyperbranched and multivalent polymers used as capping ligands [9]. Phosphines were also used as capping ligands in the early work and several phosphine-stabilized Au clusters have been reported [10,11]. Compared with phosphines, the interaction between Au and thiol is much stronger, making thiols the better capping ligands for the synthesis of Au clusters. It has been reported that Au can form a relatively strong gold−thiolate (Au−S) bond with compounds containing the thiol (−SH) or disulfide group (S−S) [12]. The formation of a gold–sulfur bond is the driving force for the anchoring of thiol ligands on gold surfaces [13]. For this reason, more recent protocols have been developed for the synthesis of Au NCs that relies on thiols rather than phosphines. Thiol-containing molecules are the most commonly adopted stabilizers in Au NC synthesis owing to the strong interaction between thiols and Au [14,15,16,17]. Thiolate-protected gold clusters, usually termed Au_n_(SR)_m_, possess a strong bond between metal and thiolate and are, therefore, very stable against degradation [18,19]. Motivated by their potential applications, proteins, as natural biopolymers, have certain structures, functional groups, and spaces that can be utilized as templates for preparing fluorescent clusters. Xie and coworkers have developed a “green” approach to synthesizing fluorescent Au clusters by employing proteins as capping ligands in order to prepare protein-based Au NCs [20]. One advantage of this method is that the three-dimensional complex structures of the proteins can be conjugated with other systems. For instance, a folic-acid-conjugated Au NC was synthesized and used as a carrier to be fused with cells via the interaction of folic acid to its receptors on the cells [21]. Therefore, capping ligands plays a key role in applications. In the past few years, a variety of proteins have also been used as capping ligands for the synthesis of Au NCs [22,23,24]. It is conceivable that SH-containing cysteines could play an important role in the synthesis of protein-protected Au NCs.

Molecules with disulfide bonds are yet to be utilized as the capping ligands for synthesizing Au NCs. Inspired by the fact that BSA contains 17 disulfide bonds and only one thiol group, we utilized quasi-spherical Ag NPs in this work to serve as a platform that plays a dual function in Au NC synthesis, both as a reducing agent for Au (III), and for attracting capping ligands to the surface of Ag NPs. The new method in this study was demonstrated to synthesize fluorescent Au NCs in mild conditions with two different molecules: the oxidized form of GSSG, and the reduced form of DTT.

## 2. Materials and Methods

### 2.1. Materials 

Silver nitrate, sodium citrate, KI, KCl, glutathione disulfide in the oxidized form (GSSG), and ethidium bromide (EtBr) were all purchased from Sigma-Aldrich (Sigma Aldrich Inc., St. Louis, MO, USA). DL, 1, 4-dithiothreitol (DTT) were purchased from Acros Organics (Geel, Belgium). Chloroauric acid trihydrate was purchased from Alpha Aesar (Thermo Fisher Scientific, Ward Hill, MA, USA). None of these chemical reagents were further purified before being used. Double-distilled water (ddH_2_O) was used in all of the experiments.

### 2.2. Synthesis of Quasi-Spherical Ag NPs

The method to synthesize the quasi-spherical Ag NPs followed the procedure in a previous report, except that the UVB lamp was replaced by violet LEDs (wavelength: 405 nm, the average intensity: 40 mW/cm^2^) [25]. In general, silver nitrate solution (0.5 mL, 0.01 M) was mixed with 0.066 g sodium citrate. Double-distilled water was added to a total of 50 mL to obtain the final concentration of 0.1 mM AgNO_3_ and 4.5 mM sodium citrate. The solution was mixed by stirring for 10 min. The mixture was then irradiated under violet LEDs for an additional 2.5 h (Appendix A). 

### 2.3. Synthesis of Au NCs

In a typical 1 mL solution for thiolated Au NC synthesis, 350 µL of ddH_2_O, 480 µL of quasi-spherical Ag NPs, 20 µL of HAuCl_4_, and 150 µL of GSSG (or DTT) were sequentially placed in a vial, and mixed. The final concentration of HAuCl_4_ and GSSG (or DTT) were 0.4 mM and 15 mM, respectively. The total silver concentration was 4.8 × 10^−2^ mM. The mixture was then incubated at 37 °C for 7 h. The reaction solution was monitored under a UV-box and a fluorescent spectrophotometer. The removal of silver ions may be necessary if any of their effects in follow-up experiments are a concern.

### 2.4. Fluorescence Spectroscopy

All of the fluorescence measurements were carried out by a Fluorolog^®®^-3 spectrophotometer (HORIBA Jobin Yvon Inc., Edison, NJ, USA). Typically, 250 µL of a fluorescent sample was placed into a 4 mm × 4 mm quartz cuvette for fluorescence measurement. Blank (ddH_2_O) subtraction was used for each measurement. The sample temperature was maintained at room temperature. The excitation shutter was kept closed, except during measurements, in order to minimize photobleaching, if any. The excitation and emission slits were opened at 5 nm and 5 nm, respectively.

### 2.5. TEM Imaging of Ag NPs and Au NCs

A Joel JEM-2100 (Japan Electron Optics Laboratory Co., Ltd., Japan) transmission electron microscope (TEM) was employed to obtain TEM images of Ag NPs and Au NCs. The TEM instrument was operated at 100 kV and 200 kV. Before being analyzed by the TEM, the Ag NP colloid was dripped onto a carbon-coated copper grid and air-dried at room temperature. All UV-Vis extinction spectra were recorded at 25 °C on a Hitachi U-5100 spectrophotometer (Hitachi Science & Technology, Japan) or a high-speed UV-Vis spectrometer (Flame, Ocean Optics, FL, USA), connected with the light source (DH-2000-BAL, Ocean Optics, FL, USA), using a quartz cuvette with an optical path of 1.0 cm.

### 2.6. Quantum Yields

A quantum yield reflects the efficiency of a given fluorophore. It is defined as follows:

Number of protons emitted/number of protons absorbed, or
Q = Q_R_(*I* × OD_R_ × n^2^)/(*I*_R_ × OD × n_R_^2^) (1)
where Q is the sample’s quantum yield; *I* is the sample’s total fluorescent intensity; OD and OD_R_ are the optical densities at the absorption (or excitation) wavelength for the sample and the reference, respectively; n is the refraction index of the solution containing the sample; Q_R_ is the standard’s quantum yield (ethidium bromide, 20%); *I*_R_ is the standard’s total fluorescent intensity (ethidium bromide); n_R_ is the refraction index of the standard in the solvent. The values of n and n_R_ are the same since both solvents are water. After integrating the emission intensities and optical densities at defined wavelengths, the quantum yields of Au NCs can be calculated by Equation (1).

### 2.7. Collisional Quenching by Iodide Ions and Other Quenchers

Four titrant solutions were prepared, respectively. They were: 1 M KI; 0.667 M KI with 0.333 M KCl; 0.333 M KI with 0.667 M KCl; and 1 M KCl. The Au NC solutions (250 µL) were each placed in 4-quartz microcells. After the initial fluorescence intensity was measured, an equal volume of (16 µL) of 1 M KI, 0.667 M KI with 0.333 M KCl, 0.333 M KI with 0.667 M KCl, or 1 M KCl was added into each cuvette, respectively. The final concentrations of KI were 60 mM (0 mM KCl), 40 mM (20 mM KCl), 20 (40 mM KCl), and 0 mM (60 mM KCl), respectively, in each cuvette. The fluorescence intensity of each sample was then measured.

The Stern-Volmer equation, shown below, was employed to analyze the collisional quenching of the samples by iodide ions:(F_0_/F) − 1 = K_SV_ [I^−^] (2)
where the sample titrated with 60 mM KCl was designated F_0_, and the fluorescence intensities of the other three cuvettes were designated F. The Ksv is the Stern-Volmer quenching constant. The value of Ksv is determined by measuring the slope of the graph (F_0_/F) vs. [I^−^].

### 2.8. Electrophoresis

To prepare, 2% agarose gel (0.50 g of agarose) was suspended in 25 mL of 1X TAE buffer [40 mM Tris-acetate (pH 7.8) and 1 mM EDTA (pH 8.00)], and a microwave was used to heat the mixture. The agarose was poured into a small gel holder after it had cooled down to about 40 °C. The agarose was then allowed to cool to room temperature for an hour. The product solution (10 µL) was mixed with 1 µL of 30% glycerol before loading to the 2% agarose gel. The sample was run at a constant voltage of 90 volts in 1X TAE buffer for around 30 min. The gel was examined under UV light to detect the location of the luminance bands.

## 3. Results

### 3.1. Synthesis of Quasi-Spherical Silver Nanoparticles (Ag NPs)

The silver nanoparticles were prepared before the Au NCs were synthesized. A UV-Visible spectrophotometer was employed to detect whether the surface plasmon resonance (SPR) absorption band was formed at around 400 nm, an indication that the quasi-spherical silver nanoparticles were synthesized. A yellow solution was observed after 2 h of LED irradiation on the synthesis mixture, indicating that quasi-spherical silver nanoparticles were synthesized. Under UV-visible spectrophotometer detection, the maximal absorption of the product was detected at 415 nm (Figure 1a). The peak intensity was not enhanced when longer LED irradiation was applied. The synthesized quasi-spherical Ag NPs were further verified by the detection by TEM. Quasi-spherical Ag NPs are shown in the TEM images (Figure 1b). The particle sizes were randomly distributed, and the average size was approximately 30 nm. The colloid Ag NP solution was then stored at room temperature for subsequent use.

### 3.2. Synthesis of Gold Nanoclusters Using GSSG as Capping Ligands

The as-prepared Ag NPs were used in the synthesis of Au NCs. Three major components are required to be present in the Au NC synthesis solution: Au^3+^, reductants, and capping ligands. In the experiment, Ag NPs were used as reductants, and oxidized glutathione (GSSG) was used as the capping ligand for the synthesis of Au NCs. The oxidized GSH was used as a capping ligand because of its peptide-like structure, composed of glutamic acid, cysteine, and glycine. In the typical synthesis of Au NCs, quasi-spherical Ag NPs, HAuCl_4_, and GSSG were sequentially placed in the reaction solution and mixed. The mixture was then incubated at 37 °C. The reaction solution gradually turned into a brilliant red color under the detection of the UV-box (Figure 2a). In the absence of Ag NPs, no fluorescent emission was detected, which revealed that Ag NPs were required to produce Au NCs in the solution (Appendix A). In addition, the reaction solution turned into a purple-red color in the absence of the capping ligand, GSSG. The absorption peak, at 550 nm, was gradually detected approximately 2 h after the Ag NPs were mixed with HAuCl_4_ (Appendix A), an indication that the reduced Au atoms aggregated and that nanoparticles were formed in the absence of capping ligands. The synthesized Au NCs were characterized using fluorescence spectroscopy, quantum yield calculation, and TEM detection. The maximal fluorescent excitation and emission wavelengths were determined and located at 354 nm and 623 nm, respectively (Figure 2b).

Under the detection of TEM, particles that were 5~6 nm in size were observed (Figure 3a). The average size of the particles, as shown on the TEM images, was larger than those previously reported [2,26], although a similar size (6.3 nm) of Au NCs has also been reported, using BSA as capping ligands [27]. To verify that the images of the particles shown on the TEM are Au NCs, not the smaller size of Ag NPs, EDS was employed to determine the components of the particles. The results showed that the major components of the particles were Au, C, and S (Figure 3b). No silver element was present in the EDS analysis. These results indicate that the particles are indeed Au NCs and GSSG, and that they could be directly associated with the Au atoms.

To determine the quantum yield of the prepared Au NCs, ethidium bromide (EtBr) was used as the parallel standard sample in an aqueous solution because of its similarity, both in excitation and emission wavelengths. The quantum yield of the synthesized Au NCs in this method was found to be 7.3%.

Au NCs were prepared under the aqueous solution. No precipitation was observed after the synthesis was completed, an indication that the synthesized Au NCs were soluble in aqueous solution. It was presumed that the glutathione-Au NCs are accessible to the aqueous environment. However, the Au NCs are associated with the capping ligands, GSSG. The capping ligands might fabricate a protective layer to the aqueous environment. One approach to address this issue is to determine whether the capping ligands block or reduce the exposure of the fluorescent Au NCs to the solvent (water). Water-soluble iodide ions were used to detect whether they can quench the fluorescence intensity of the thiolated Au NCs. When iodide ions, or certain molecules, collide with an excited fluorescent dye, they return the fluorophore to its ground state without the emission of a photon. As a result, the original fluorescent intensity emitted by the sample is quenched, i.e., the intensity is reduced. The extent of the quenching depends on the frequency of the collisions and, hence, is directly proportional to the concentration of quencher that has access to the fluorophore, as expressed in the Stern-Volmer equation. Thus, the sensitivity of Au NCs to collisional quenching by iodide ions can be used to evaluate the access of the signal sequence to the aqueous solvent containing iodide ions. This approach has been used previously to examine and assess the accessibility of fluorescent probes in complexes [28]. When the glutathione-Au NCs were titrated with water-soluble iodide ions, the higher iodide concentration meant lower fluorescent intensities (Figure 4a). KCl was also added to compensate for the shortage of ionic strength for the lower iodide concentration to ensure that the ionic strength did not interfere with the fluorescent alteration (Figure 4a). The Stern-Volmer constant (K_SV_) was calculated to be approximately 26.2 M^−1^ (Figure 4b). Ksv is a measure of the fluorophore accessibility to its environment, i.e., it is an indication of how many of the Au NCs are protected by GSSG: a higher K_SV_ value indicates that the fluorophore is more accessible to the water environment. The high K_SV_ value of glutathione-Au NCs implies that the fluorescent probes in the Au NCs are exposed to the aqueous solution.

### 3.3. The Role of Ag NPs in the Synthesis of Au NCs

The reduction potential of Ag^+^ and Au^3+^ is 0.8 V and 1.40 V, respectively. Ag NPs are reasonably reduced to Au^3+^. To monitor the presence of quasi-spherical Ag NPs during the synthesis of Au NCs, UV-Visible spectroscopy was used to detect the characteristic absorption of Ag NPs (Figure 5a). The peak, at 415 nm, had almost disappeared immediately after all of the reaction components were mixed. The reaction solution turned to colorless from light yellow after the synthesis was completed. No purple-red color was observed, implying that the reaction did not produce detectable Au nanoparticles. This result indicated that the quasi-spherical Ag NPs could be oxidized to Ag^+^, and that the Au^3+^ ions were reduced to Au atoms, or Au^+^, during the formation of the Au NCs. To confirm that Ag NPs played the reductant role in the synthesis of Au NCs, Ag NPs and HAuCl_4_ were exclusively placed in the solution without GSSG. A purple color gradually appeared in the solution approximately 90 min after the reaction Appendix A, indicating that the Au^3+^ ions were reduced and aggregated to the Au nanoparticles by the reductant Ag NPs in the absence of GSSG. Therefore, the GSSG molecules played the role of the protecting molecules during Au NC synthesis, and prevented the reduced Au atoms from aggregating into Au nanoparticles.

To understand whether Ag NPs associated or interacted with GSSG, Ag NPs and GSSG were mixed together, and the extinction spectra were detected by the UV-Vis spectrometer. The intensity of the feature peak for the Ag NPs at 415 nm was greatly reduced (Figure 5b). However, the TEM images confirmed that the Ag NPs still remained in the solution (Appendix A). EDS analysis showed that the elements, Ag and S, were detected on the particles area, indicating that GSSG could be in the proximity of Ag NPs after the addition.

### 3.4. Using Dithiothreitol as Capping Ligands

To demonstrate whether this method can be generally applied on other protecting molecules for the synthesis of Au NCs, other molecules were tested for the synthesis. GSSG contains a disulfide bond, formed by two reduced thiol groups. Consequently, a molecule with two thiols without oxidation was tried for the synthesis. Dithiothreitol (DTT), containing two thiols with a small molecular weight, was selected. The results showed that DTT was able to synthesize DTT-Au NCs using the same approach. The quantum yield of DTT Au NCs was found to be 7.0%, very similar to that of the glutathione-Au NCs. Interestingly, the iodide collisional quenching experiment showed that the K_SV_ of the DTT-protected Au NCs is close to 0, indicating that the fluorophore in the clusters is not accessible to water (Figure 4b, red line).

### 3.5. Electrophoretic Mobiliy of Glutathione-Protected and DTT-Protected Au NCs

Two different Au NCs have been synthesized: glutathione-Au NCs and DTT-Au NCs. One primary structural difference is on the charge. GSSG bears two negative charges, while DTT does not have any charge under neutral conditions. However, the total net charge of the thiolated Au NCs is not clear. We examined whether the protecting group influences the electrophoretic mobility of the fluorophore of the Au NCs. As shown by the results, the glutathione-protected Au NCs moved toward the anode under a pH 7.8 buffer, indicating that they contain a negative charge (Figure 6). By contrast, the DTT-protected Au NCs moved neither forward nor backward, and almost stayed in the original place. These results demonstrate that the charge on the protecting molecules dictates the electrophoretic mobility of the synthesized Au NCs and reveals that the protecting molecules are associated with the fluorophore. These results also suggest that cation exchange resin, packed in a small syringe, can be used to absorb and remove Ag^+^ from the product suspension, if necessary.

## 4. Conclusions

In this study, a facile synthesis method for thiolated Au NCs has been developed. The synthesis of Au NCs was completed within 7 h at 37 °C, when quasi-spherical Ag NPs were available. In the past, one of the troubling matters has been choosing the protecting groups, usually huge polymers with branched skeletal structures. In this synthesis, much smaller molecules, with disulfide bonds or two thiols, were able to be used as protecting ligands. Quasi-spherical Ag NPs were used in the synthesis and played two key roles: to act as a reducing agent, and to serve as a platform to accumulate protecting molecules on the surface of the Ag NPs (Figure 7). Au ions on the surface of the quasi-spherical Ag NPs were trapped and reduced by the Ag NPs, while the protecting molecules were attracted to the surface of the Ag NPs, resulting in an association with reduced Au ions. The quantum yields of GSSG and DTT Au NCs are 7.3% and 7.0%, respectively. An electrophoretic mobility assay showed that the charge on the protecting molecules could dictate the total charges on the synthesized Au NCs. One advantage of this method is that the protecting group could be variable. We have demonstrated that molecules, either with two thiols or reduced thiols, were able to be used as the protection molecules. GSSG is the oxidized form of two GSHs, consisting of three amino acids, glutamic acid, cysteine, and glycine. It is conceivable that short peptides, containing two thiols or reduced thiols, can be used as the protecting molecule. Our results demonstrate that the Ag NP-mediated method could become an easy approach for synthesizing thiolated Au NCs with different capping molecules. In addition, the morphology of the Ag NPs influences their ability to attract the ligands to the surface. Consequently, the shape and distribution pattern of Ag nanoparticles are critical for the synthesis process. However, more experiments need to be carried out to confirm the versatility of this method.

## Figures and Tables

**Figure 1 nanomaterials-11-02835-f001:**
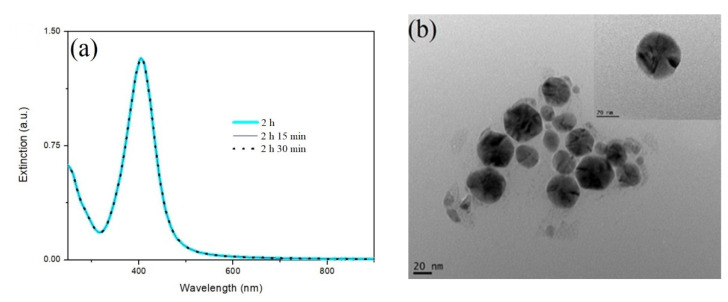
(**a**) UV-Vis spectra and (**b**) TEM images of quasi-spherical Ag NPs synthesized by LED irradiation. The LSPR band of the quasi-spherical Ag NP colloids peaked at 415 nm, and the intensity was not further increased for longer irradiation.

**Figure 2 nanomaterials-11-02835-f002:**
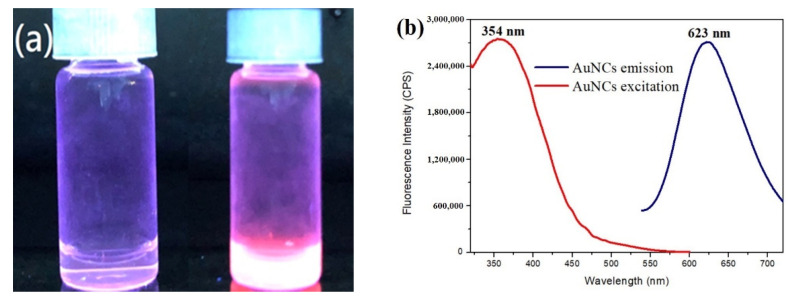
(**a**) The image of aqueous solutions of newly synthesized glutathione-Au NCs under UV-box detection. Two experiments were carried out, in the presence (right), and absence (left) of Ag NPs, respectively. A bright red color was observed in the solution on the right bottle, and the solution in the left bottle was colorless. (**b**) The excitation (red line) and emission (black line) spectra of the synthesized Au NCs.

**Figure 3 nanomaterials-11-02835-f003:**
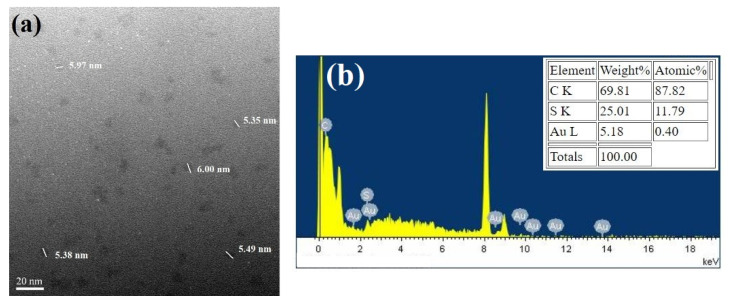
(**a**) TEM image of Au NCs. The sizes of the particles are between 5~6 nm. (**b**) Energy dispersive X-ray spectrometer (EDS) analysis results: the particles are comprised of C, S, and Au elements (upper panel). No Ag element was detected.

**Figure 4 nanomaterials-11-02835-f004:**
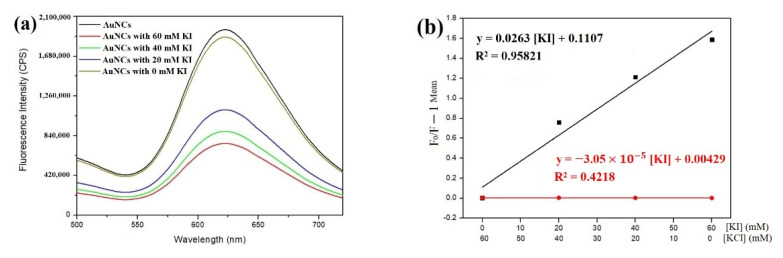
Iodide ions quenching glutathione-Au NCs. (**a**) The emission intensity of glutathione-Au NCs was quenched by the addition of iodide ions (KI). Chloride (KCl) was added to ensure that each sample had the same ionic strength, even though the concentration of iodide differed. Concentrations of total ions {[I^−^] + [Cl^−^]} are 60 mM. (**b**) The net fluorescence intensity obtained from each sample, which was titrated with 60 mM KCl, was designated F_0_, and the intensities of the other three samples were designated F. glutathione-Au NCs (black line), and DTT-Au NCs (red line).

**Figure 5 nanomaterials-11-02835-f005:**
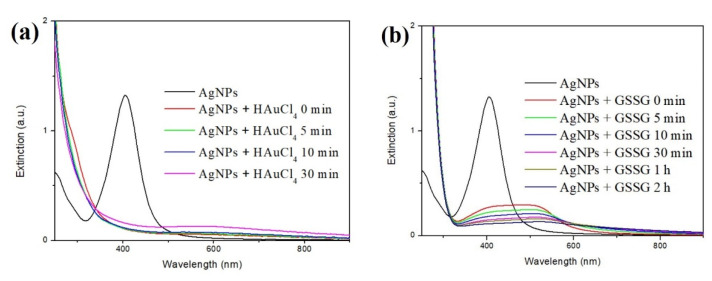
(**a**) Time course detection of UV-Visible spectrophotometer for the Au NCs synthesis. (**b**) Time course detection of UV-Visible spectrophotometer for the association of GSSG with Ag NPs.

**Figure 6 nanomaterials-11-02835-f006:**
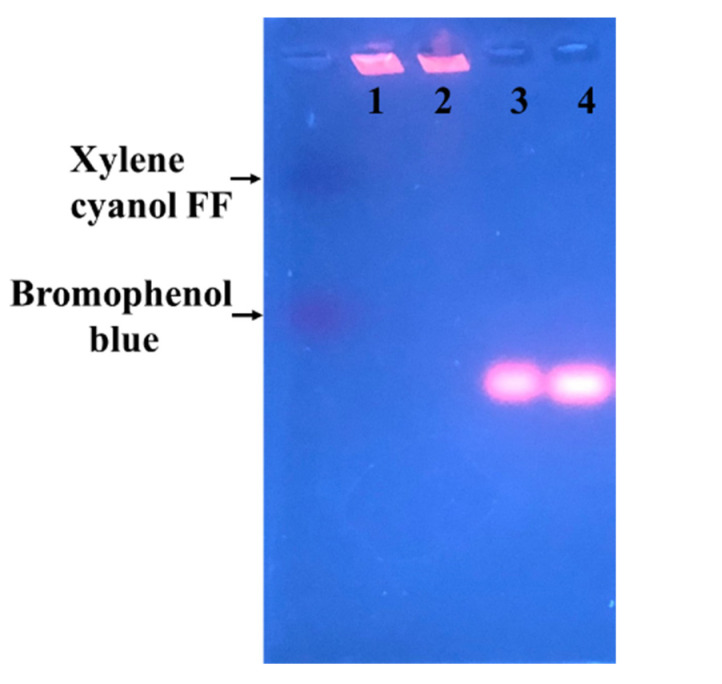
Electrophoretic mobility of Au NCs on 2% agarose gel. The samples on Lanes 1 and 2 are DTT-Au NCs. Lanes 3 and 4 are glutathione-Au NCs. Bromophenol blue and xylene cyanol FF, as references, are indicated.

**Figure 7 nanomaterials-11-02835-f007:**
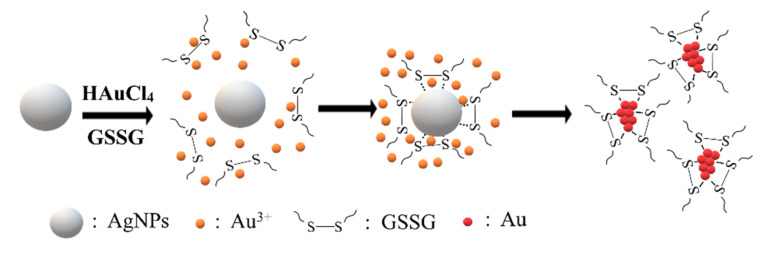
Ag NPs serve as a platform to synthesize thiolated Au NCs.

## Data Availability

All data generated and analyzed during this study are included in this paper and the attached Appendix A.

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
