# Peer review of "Silver Nanoparticle-Mediated Synthesis of Fluorescent Thiolated Gold Nanoclusters"

_nanomaterials, 2021, doi:10.3390/nano11112835_

Round 1
Reviewer 1 Report
Manuscript entitled Silver Nanoparticles-Mediated Synthesis of Fluorescent Thiolated Gold Nanoclusters by Cheng-Yeh Chang describes an approach for gold nanoclusters synthesis using silver nanoparticles.
In my opinion the manuscript has several major issues that have to be addressed before it can be considered for publication.
1) The significance is not sufficiently highlighted. It is not clear why the reader should prefer the presented method over other existing one.
2) It is not clear, how Ag (either ions or NPs) was removed from the final product
3) Authors state that Au NCs are coated by GSSG, however I believe that it is reduced within the solution to GSH and the actual capping agent is GSH via the SH group.
Minor comments:
1) The quality of figures should be improved (the trace colors are very similar, Fig. 7 is almost invisible,..)
2) The term “electromobility” should be replaced by “electrophoretic mobility” this value should be calculated
3) the molecular mass ladder should be added in figure
4) excitation and emission maxima should be specified
5) The sentence: “..solution was detected by UV-box…” should be rephrased. Nothing can be detected by UV-box.
Reviewer 2 Report
Nanoparticles of noble metals are of scientific interest as promising in molecular electronics, sensors, and biomedical diagnostics. With decreasing particle size, changes at the electronic level that affect electrical conductivity depend on the chemical environment and are most pronounced between 1 and 3 nm. This corresponds to gold clusters. The intrinsic properties of clusters, such as fluorescence, can be made available for bionanotechnological applications by linking them to biomolecules. An important point in the synthesis of nanoclusters is the preservation of the stability of the obtained nanoparticles or preventing their aggregation. The aim of the project is to synthesize fluorescent Au nanoclusters using thiols for stabilizing.
The novelty of the synthesis is in using of Ag nanoparticles as a reducing agent for Au (III) and for attracting the preventing molecules. Two types of molecules differ in charge were compared in the study, namely, glutathione disulfide and dithiothreitol, in oxidized and reduced forms, respectively.
Since the importance of Ag nanoparticles in the synthesis it is not enough information concerning the nanoparicles. How critical are the shape and distribution pattern of Ag nanoparticles?
There is no IR in the equation (1).
Round 2
Reviewer 1 Report
Authors should quantify Ag remaining within the resulting suspension (e.g. ICP-MS) and include some discussion commenting the requirement of its removal from the final product before use. Also the purification procedure shoud be suggested.
